# DAFuzz: data-aware fuzzing of in-memory data stores

Yingpei Zeng[1], Fengming Zhu[1], Siyi Zhang[1], Yu Yang[1], Siyu Yi[1],
Yufan Pan[1], Guojie Xie[2] and Ting Wu[3]

[1] School of Cyberspace, Hangzhou Dianzi University, Hangzhou, China
[2] Zhejiang Key Laboratory of Open Data, Hangzhou, China
[3] Hangzhou Innovation Institute, Beihang University, Hangzhou, China



## ABSTRACT

Fuzzing has become an important method for finding vulnerabilities in software. For fuzzing programs expecting structural inputs, syntactic- and semantic-aware fuzzing approaches have been particularly proposed. However, they still cannot fuzz in-memory data stores sufficiently, since some code paths are only executed when the required data are available. In this article, we propose a data-aware fuzzing method, DAFuzz, which is designed by considering the data used during fuzzing. Specifically, to ensure different data-sensitive code paths are exercised, DAFuzz first loads different kinds of data into the stores before feeding fuzzing inputs. Then, when generating inputs, DAFuzz ensures the generated inputs are not only syntactically and semantically valid but also use the data correctly. We implement a prototype of DAFuzz based on Superion and use it to fuzz `Redis` and `Memcached`. Experiments show that DAFuzz covers 13~95% more edges than AFL, Superion, AFL++, and AFLNET, and discovers vulnerabilities over 2.7× faster. In total, we discovered four new vulnerabilities in `Redis` and `Memcached`. All the vulnerabilities were reported to developers and have been acknowledged and fixed.

# INTRODUCTION

Fuzzing has become an important way to find vulnerabilities in software (*Manes et al., 2019*; *Zhu et al., 2022*), and coverage-guided fuzzing (CGF) (*Zalewski, 2017*; *Böhme, Pham & Roychoudhury, 2016*) is one of the most popular fuzzing technologies, since it could gradually explore the state space of the program under test (PUT) even if only several initial seeds are given. This is because when it mutates seeds to create new inputs, it traces the coverage information of the new inputs, and adds the inputs into the seed pool as new seeds if the inputs have new code coverage. CGF fuzzers (*e.g.*, AFL (*Zalewski, 2017*), honggfuzz (*Honggfuzz, 2023*), libFuzzer (*libFuzzer, 2023*), and AFL++ (*Fioraldi et al., 2020*)) are the main fuzzers used in the famous OSS-Fuzz project (*Google Security Team, 2018*), which has discovered over 8,900 vulnerabilities and 28,000 bugs across 850 open-source projects by February 2023 (https://github.com/google/oss-fuzz). CGF fuzzers have also been used to discover vulnerabilities in other fields including operation systems (*Google, 2015*; *Pan et al., 2021*), network protocols (*Pham, Böhme & Roychoudhury, 2020*; *Ba et al., 2022*), as well as the Internet of Things (IoT) (*Zheng et al., 2019, 2020*).

Corresponding author
Yingpei Zeng, yzeng@hdu.edu.cn

| | |
|---|---|
| **Data aware** | e.g., with required data |
| Semantics aware | e.g., defined variables |
| Syntax aware | e.g., correct formats |

**Figure 1 DAFuzz considers the syntax, semantics, and data (which is a newly proposed factor) at the same time in fuzzing.**

It is known to be difficult for CGF fuzzers to fuzz programs expecting structural inputs because it is hard to get syntactically and semantically valid inputs through seed mutation (*Wang et al., 2019*). For instance, common mutation operations like bit flipping and byte modification (*Zalewski, 2017*; *Fioraldi et al., 2020*) can often corrupt the original format of an input. To tackle the problem, grammar-aware CGF fuzzers that understand the grammar (syntax and semantics) of inputs have been proposed recently (*Wang et al., 2019*; *Han, Oh & Cha, 2019*; *Padhye et al., 2019*; *Park et al., 2020*; *He et al., 2021*; *Liang, Liu & Hu, 2022*). They usually consider syntactic and semantic constraints when creating new inputs, and then the created new inputs could pass the corresponding syntax and semantics checks in the code. For instance, when fuzzing JavaScript engines, Superion (*Wang et al., 2019*) guarantees the generation of syntactically valid JavaScript inputs, such as ensuring that brackets always exist in pairs. Additionally, SoFi (*He et al., 2021*) can further ensure the creation of semantically valid JavaScript inputs, such as ensuring variables are defined before they are used.

In-memory data stores like `Redis` are widely used in thousands of companies like Twitter and Snapchat (https://redis.io/docs/about/users/), since they provide very efficient and convenient access to data. It is crucial to uncover any vulnerabilities in them in a timely manner. However, when applying CGF fuzzers to fuzz in-memory data stores, merely considering the syntactic and semantic validity of the inputs is insufficient. In-memory data stores usually use different commands with defined parameter formats to access the data, and may use serialization protocols for communicating between their clients and servers. However, even if commands are sent in syntactically and semantically format, some code paths may not be executed when the required data are not available. This is because the statuses of the data directly control how commands are processed and which code paths are executed. For example, the processing of command "RPOP *key*" in `Redis` may execute a quick-exit code path and return an empty array when the list *key* is empty, and may execute a popping data code path and return the last element of the list only when the list *key* is not empty.

In this article, we propose DAFuzz, a data-aware fuzzing method for in-memory data stores that considers data requirements in addition to syntactic and semantic validity in fuzzing, as shown in Fig. 1. We also compare DAFuzz with other typical CGF fuzzers in Table 1. Although general CGF fuzzers like AFL (*Zalewski, 2017*) and AFL++ (*Fioraldi et al., 2020*) can be applied to fuzz in-memory data stores, they are not grammar-aware.

**Table 1 Conceptual comparison of typical CGF fuzzers.**

| Fuzzer | Syntax-aware | Semantics-aware | Data-aware | Applied to in-memory data stores |
|---|---|---|---|---|
| AFL (*Zalewski, 2017*), AFL++ (*Fioraldi et al., 2020*) | No | No | No | Yes |
| AFLNET (*Pham, Böhme & Roychoudhury, 2020*) | Partial | No | No | Yes |
| Superion (*Wang et al., 2019*) | Yes | No | No | Yes |
| SoFi (*He et al., 2021*) | Yes | Yes | No | No (JavaScript engine only) |
| Squirrel (*Zhong et al., 2020*), SQLRight (*Liang, Liu & Hu, 2022*) | Yes | Yes | No | No (SQL database only) |
| DAFuzz (this article) | Yes | Yes | Yes | Yes |

AFLNET (*Pham, Böhme & Roychoudhury, 2020*) and Superion (*Wang et al., 2019*) are partially or fully syntax-aware but not semantics-aware. Other fuzzers such as SoFi (*He et al., 2021*), Squirrel (*Zhong et al., 2020*), and SQLRight (*Liang, Liu & Hu, 2022*) are both syntax-aware and semantics-aware, but they are specifically designed for and limited to fuzzing JavaScript engines or SQL databases. Also, all these fuzzers are not data-ware. In contrast, DAFuzz proposed in this article is syntax-aware, semantics-aware, and data-aware, and is designed for fuzzing in-memory data stores. Specifically, firstly, DAFuzz designs a data construction algorithm to generate a special data set that is used to satisfy the requirements of different code paths. Then, DAFuzz generates inputs according to the data set, as well as the grammar of commands. Finally, DAFuzz also uses syntax-aware mutation to improve mutation efficiency. We implement a prototype of DAFuzz based on Superion and test DAFuzz with two popular in-memory data stores `Redis` and `Memcached`. The experiments show that DAFuzz could find 13~95% more edges than AFL, Superion, AFL++, and AFLNET in 24 h, and discover the same number of edges at least 26×, 21×, 19×, and 288× faster than AFL, Superion, AFL++, and AFLNET, respectively. In addition, DAFuzz discovers the same vulnerabilities as other fuzzers but discovers them at least 2.7× faster. All four discovered vulnerabilities (three in `Redis` and one in `Memcached`), including three segmentation violations and one stack buffer overflow (more details in the "Vulnerabilities Discovered" section), were reported to the developers and have been acknowledged and addressed in new releases.

In summary, this article contains the following contributions.

- We propose the first data-aware fuzzing method for reaching code paths that are executed only when the required data are available.
- We design an algorithm to construct a data set to load for fuzzing, and an algorithm to generate fuzzing inputs that use the required data and are syntactically and semantically valid as well.
- We implement a prototype of DAFuzz based on Superion, and compare DAFuzz with other state-of-the-art fuzzers including AFL, Superion, AFL++, and AFLNET using two popular in-memory data stores `Redis` and `Memcached`. We discover four new vulnerabilities and report them to the developers. We share the DAFuzz prototype as

**Algorithm 1 The DAFuzz fuzzing loop.** The differences between DAFuzz and AFL are underlined.

**Input:** Initial seed set **s**, grammar *G*, data set *D*

1: **repeat**
2:     generate inputs with *G* and *D*, and fuzz them
3:     *s* = ChooseNext(**s**)
4:     deterministically fuzz with *s* if needed
5:     mutate *s* in havoc/splicing style and fuzz the mutated inputs
6:     mutate *s* in syntax-aware style with *G* and fuzz the mutated inputs
7: **until** *timeout* reached or user *aborts*

**Output:** Crash inputs **s$_c$**

Free and Open Source Software at https://github.com/hdusoftsec/DAFuzz (release after publication).

# BACKGROUND AND MOTIVATING EXAMPLE

## Background

### In-memory data store

In-memory data stores like `Redis` and `Memcached` are widely used in web and mobile application servers. They are popular because they store data in *memory* and provide very efficient and convenient access to the data, which makes them quite suitable for tasks like caching and session management. In-memory data stores may support different kinds of data types, for example, `Redis` supports *string, list, set, sorted set, hash, etc.* (https://redis. io/docs/data-types/tutorial/). They usually use different commands with defined parameter formats to access data. For example, in `Redis` "SET *key value*" is for storing *value* to *key*, and both *value* and *key* could be arbitrary strings. They may also use some serialization protocols (*i.e.*, application layer network protocols) for client-server communication (*i.e.*, for sending and receiving commands and data). For example, `Redis` mainly uses RESP (REdis Serialization Protocol) protocol (https://redis.io/docs/reference/ protocol-spec/) for client-server communication. It is important to eliminate any vulnerabilities in such widely used programs, especially since they usually run on servers that have rich computation and network resources.

### Fuzzing

Fuzzing, a technique used for vulnerability discovery, has a history of over 30 years (*Miller, Fredriksen & So, 1990*). It has been widely recognized as an effective method and can complement other techniques such as manual code inspection and static program analysis (*Godefroid, 2020*). Coverage-guided fuzzing (CGF) now probably is the most popular fuzzing technology (*Manes et al., 2019*; *Zhu et al., 2022*). CGF fuzzer usually first instruments the PUT to trace coverage information when running each input, and starts a fuzzing loop with some initial seeds. The fuzzing loop of AFL is also shown in Algorithm 1

(without the grey part). In the fuzzing loop, it keeps selecting seeds and mutating them to create new inputs and runs the inputs with the PUT. The mutation of a seed may include a deterministic stage in which the seed is sequentially changed by predefined steps like one-by-one bit flipping, and an indeterministic (havoc/splicing) stage in which the seed is applied with stacking changes (*i.e.*, applying multiple changes to produce one input) and may be spliced with another randomly selected seed first. If any new inputs have new code coverage, they are added to the seed pool as new seeds. Thus, the fuzzer could gradually explore more state space of the PUT, compared to traditional unit testing and mutation testing.

Mutation-based CGF fuzzers do not perform well for programs expecting structural inputs, such as JavaScript engines, XML parsers, *etc.*, because it is hard for the fuzzers to get syntactically and semantically valid inputs during random seed mutation (*Wang et al., 2019*). Programs expecting structural inputs usually have syntax and semantics checks early in the program execution, and inputs that are either not syntactically or semantically valid could not pass such checks to execute deep program code paths. Syntax-aware (*Wang et al., 2019*; *Padhye et al., 2019*) and semantics-aware fuzzers (*Han, Oh & Cha, 2019*; *He et al., 2021*) have been proposed to solve the problem. They understand the grammar of inputs and could obtain new inputs that are syntactically and even semantically valid. For example, they may convert seeds into abstract syntax trees (ASTs) and mutate at the AST tree node level instead of the byte level of seeds (*Wang et al., 2019*).

### Motivating example

Only syntax-aware and semantics-aware fuzzing is not enough for efficiently fuzzing in-memory data stores like `Redis`, since some code paths could only be executed when the required data are available. Take the processing of RPOP *key* command in `Redis` for example. The command is to remove and return the last element of the *key* list, and its corresponding code snippet in `t_list.c` is shown in Fig. 2A. An input "RPOP *list1*" is both syntactically valid (*i.e.*, command format) and semantically valid (*i.e.*, using *list1* without declaration first is correct in `Redis`), however, if *list1* is empty, the execution exits at line 5 and the left code lines in the function are not executed. Another example is the processing of SINTER *key [key…]* command. The command is to return the intersection of all the given sets, and its code snippet in `t_set.c` is shown in Fig. 2B. The execution would end early at line 6 if any set is empty, and even if they are all not empty, the code lines represented between line 16 and line 18 are not executed if the intersection of the sets is empty.

## DAFUZZ APPROACH

### Overview

DAFuzz incorporates data-aware fuzzing, in addition to syntax-aware and semantics-aware fuzzing. The architecture of DAFuzz is shown in Fig. 3, and its differences from other fuzzers like AFL are highlighted. First, DAFuzz uses a data construction module to produce the data that would be used later in program execution and input generation in the fuzzing loop (the "Data Construction" section). Second, in the fuzzing loop, DAFuzz

```
1. void popGenericCommand(client *c, int where) {
2.      …
3.      robj *o = lookupKeyWriteOrReply(c, c->argv[1],
   shared.null[c->resp]);
4.      if (o == NULL || checkType(c, o, OBJ_LIST))
5.          return; // return if no data in list
6.      …
7.      if (!count) {
8.          value = listTypePop(o,where);
9.          serverAssert(value != NULL);
10.          addReplyBulk(c,value);
11.          decrRefCount(value);
12.          listElementsRemoved(c,c-
   >argv[1],where,o,1,NULL);
13.      }
14.      …
15. }
```

A

```
1.     void sinterGenericCommand(client *c, robj **setkeys, …) {
2.         /* Check empty set */
3.         …
4.         if (empty > 0) {
5.             …
6.             return;
7.         }
8.         /* Compute the intersection set */
9.         …
10.        if (cardinality_only) {
11.            addReplyLongLong(c,cardinality);
12.        } else if (dstkey) {
13.            /* Store the resulting set into the target, if the
14.             * intersection is not an empty set. */
15.            if (setTypeSize(dstset) > 0) {
16.                setKey(c,c->db,dstkey,dstset,0);
17.                addReplyLongLong(c,setTypeSize(dstset));
18.                …
19.            } else {
20.                …
21.            }
22.            …
23.        }
24.        …
25. }
```

B

**Figure 2 Code snippets for processing two commands in** Redis. (A) The RPOP command. (B) The SINTER command.                                             

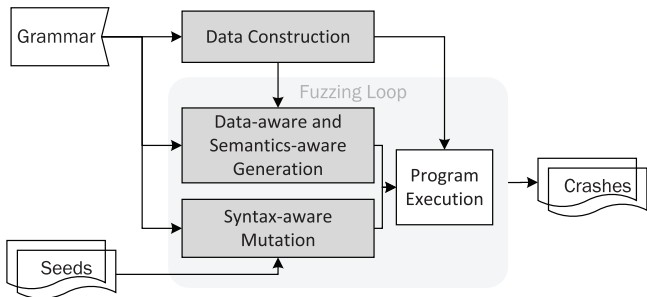

**Figure 3 The DAFuzz architecture, with differences from AFL highlighted.**
                                             

---

**Algorithm 2** Data construction for fuzzing.

**Input:** a set of data types $T$, number of data to create for each type $N$, maximum number of values inside each data $N_f$

1:   $D = \phi$                                                   ▷ The data to output

2:   $V = \phi$

3:   add predefined different values to $V$

4:   **for** each data type $t$ in $T$ **do**

5:       **for** $i$ from 1 to $N$ **do**

6:           create data item $d$ of type $t$

7:           $n_f = \text{UniformRandom}(1, N_f)$

8:           **for** $j$ from 1 to $|V|$ **do**

9:              add the corresponding value of $V$ to $d$

10:          **end for**

11:          **for** $j$ from $|V| + 1$ to $n_f$ **do**

12:             add a random string value to $d$

13:          **end for**

14:          add $d$ to $D$

15:       **end for**

16:   **end for**

**Output:** data set $D$

---

uses a data-aware and semantics-aware generation module for generating inputs that are syntactically and semantically valid, as well as referring to valid data (the "Data-aware and Semantics-aware Input Generation" section). Last but not least, DAFuzz uses syntax-aware mutation in the fuzzing loop for creating syntactically valid inputs (the "Syntax-aware Mutation" section).

The fuzzing loop of DAFuzz is shown in Algorithm 1, and its differences from AFL are highlighted as well. DAFuzz is still a CGF fuzzer like AFL, *i.e.*, with a fuzzing loop that keeps choosing a seed, mutating it to get inputs, and fuzzing inputs by feeding them to the PUT. However, in the fuzzing loop, it uses input generation as well for using the grammar and the data constructed. Using input generation together with seed mutation to create inputs is similar to some CGF fuzzers like syzkaller (*Google, 2015*). In addition to the havoc/splicing stage, DAFuzz contains a syntax-aware mutation stage like Superion (*Wang et al., 2019*) for improving the ratio of valid inputs obtained from seed mutation.

## Data construction

The data construction module is to construct a data set containing different types of data, which is later provided to program execution and input generation. The algorithm is shown in Algorithm 2. Users could specify the set of data types $T$ to generate (*e.g.*, *list* and *hash*), the number of data items $N$ to create for each type, and the maximum number of

values $N_f$ to store in each data item (*e.g.*, the number of members in *list*). In the algorithm, a small set of predefined values $V$ is prepared first. Then, for each data type, $N$ data items would be generated. For generating each data item, its number of members $n_f$ is randomly obtained by UniformRandom$(1, N_f)$ (uniformly selected in $[1, N_f]$). After that, both the predefined values $V$ and some randomly generated strings are added to the data item, according to $n_f$. The predefined value set $V$ is useful since it makes sure that the data items have common values, which makes the calculations (*e.g.*, the intersection) among them may not empty.

To run the algorithm, users should know the supported data types of the data store (PUT), and they usually should add all supported data types into $T$ to construct different types of data, unless users just want to focus on fuzzing part of data types like in directed fuzzing (*Böhme et al., 2017*). In addition, however, it is better to specify moderate values for $N$ and $N_f$, which define how much data to generate, because loading too much data when starting the server would slow down the fuzzing execution speed.

## Data-aware and semantics-aware input generation

The data-aware and semantics-aware input generation module is used to generate inputs according to the grammar of inputs and the data constructed in the previous section, and its algorithm is shown in Algorithm 3. For in-memory data stores, the grammar of inputs is mainly the grammar of commands, which consists of the name, options, and parameters of each command. In addition to the grammar and data set, DAFuzz also prepares a command list $C$ containing all commands and an optional command-to-related-commands map $R$. The map $R$ is a map that maps each command to a command list containing all its related commands, *e.g.*, the related commands of *SINTER* command (*set* intersection command) including all commands about *set* calculations. The normal distribution $N(\mu, \sigma^2)$ is to define how many commands to put inside a single input. DAFuzz does not use uniform distribution here for having a small probability to generate extraordinarily big inputs.

In the input generation algorithm, the number of commands $n_c$ in the input is first calculated. After that, the first command type is randomly selected from the command list $C$. Then, DAFuzz creates a command $c$ of the given command type, with options of the command randomly enabled. The creation method is introduced later with examples. DAFuzz fills all fields of the command $c$ before appending it to the seed. When filling a field of a data type, DAFuzz first tries to randomly select a data item from all the data items with the same data type in data set $D$. If no such data items exist in the data set, it randomly generates a data item with the given type. When selecting the next command type, if the current command has related commands in the map $R$, it obtains all related commands with $R[next\_cmd]$ and randomly selects the next command. Otherwise, it still randomly selects a command type from the whole command list as the next command. Now the algorithm does not try to ensure the "normal" order of generated commands, because it is hard to define the "normal" order (*e.g.*, hard to know which list command, LPOP or LPUSH, is normally executed first), and randomly executing commands may help to expose vulnerabilities in data stores.

---

**Algorithm 3** Data-aware and semantics-aware input generation.

**Input:** command grammar $G$, command list $C$, command-to-related-commands map $R$, data set $D$, a normal distribution $N(\mu, \sigma^2)$ deciding the number of commands in an input.

1:  $s = \phi$                                                                                      ▷ The seed to output

2:  $n_c = N(\mu, \sigma^2)$

3:  $next\_cmd = \text{SelectRandomCommand}(C)$

4:  **for** $i$ from 1 to $n_c$ **do**

5:      create command $c$ of type $next\_cmd$ according to $G$

6:      **for** each field $f$ of data type $t$ in $c$ **do**

7:          **if** $t$ exists in $D$ **then**

8:              select a random data item in $D$ of type $t$ and fill the field $f$

9:          **else**

10:             generate random data item and fill the field $f$

11:         **end if**

12:     **end for**

13:     append $c$ to $s$

14:     **if** $R[next\_cmd]$ is not empty **then**

15:         $next\_cmd = \text{SelectRandomCommand}(R[next\_cmd])$

16:     **else**

17:         $next\_cmd = \text{SelectRandomCommand}(C)$

18:     **end if**

19: **end for**

**Output:** seed $s$

---

We use examples to illustrate the aforementioned command creation, mainly about how options are enabled. Suppose the LPOP *key [count]* command is selected for creation (where "[]" means inner content is optional). DAFuzz randomly selects one from the two possible commands to create, LPOP *key* and LPOP *key count*, which means there are two possible values for the component "*[count]*": "*count*" or " " (*i.e.*, blank). Considering another ZADD command is selected and it has the grammar ZADD *key [NX|XX] [GT|LT] [CH] [INCR] score member [score member]* (where "|" means any one of the listed elements is allowed). We can see that there are three possible values for the component "*[NX|XX]*": "NX", "XX", or " " (*i.e.*, blank). Thus, for the ZADD command, DAFuzz randomly selects one from the $3 \times 3 \times 2 \times 2 \times 2 = 72$ possible commands to create. The creation method is implemented by processing "[]" and "|" symbols in multiple rounds until all of them are parsed, which could easily deal with the case that options are nested (*e.g.*, "[[…]…]").

### Syntax-aware mutation

DAFuzz uses the tree-based mutation method proposed in Superion (*Wang et al., 2019*) to mutate seeds, which could keep the syntax of test inputs correct. The tree-based mutation

method generally works as follows. It parses two seeds *tar* and *pro* into two abstract syntax trees first and collects all the subtrees into a set *S*. Then, it iterates all the subtrees of the AST of *tar* one by one, and for each subtree obtains a batch of new inputs, by replacing the subtree with each subtree in *S* once and serializing the mutated AST to an input. DAFuzz follows the same method to mutate seeds, however, it uses the grammar of in-memory data stores instead (*e.g.*, the grammar of commands and the RESP protocol for `Redis`).

DAFuzz also adopts the enhanced dictionary-based mutation method proposed in *Wang et al. (2019)*, which could cleverly mutate seeds by considering token boundaries (*e.g.*, not partially overwriting tokens). We add the names of commands and the names of data items in the constructed data set into the fuzzing dictionary to better support the method.

Note that DAFuzz still would mutate seeds to produce inputs that are not syntactically or semantically valid, since it still has the havoc/splicing stage as we mentioned in Algorithm 1. This is helpful to cover exception handling code during fuzzing, and we do observe a higher code coverage in our testing.

## IMPLEMENTATION

### Preparing the grammars

The grammars mainly include the grammars of commands and serialization protocols and are obtained from corresponding official websites. For example, `Redis` has more than 300 commands in 6.x, and we retrieve its command grammar (including command name, options, parameters) from https://redis.io/commands/ with a web crawler written in Python. In the document for each command, there is also a column for "Related commands", which is used to build the command-to-related-commands map. `Redis` may use both RESP and inline commands as its serialization protocol, and DAFuzz supports both of them in its syntax-aware mutation. The grammars are stored in JSON files and are provided to the fuzzer at runtime.

### Data construction

The data constructed for `Redis` are stored into a `rdb` file, which is loaded when starting the `Redis` server with the `--dbfilename` option. However, `Memcached` does not have a mechanism to load data into the data store when starting the server, and the constructed data are loaded by using commands that are inserted in the front of the initial seeds. The more data are constructed for fuzzing, the higher probability that different data conditions would be satisfied during fuzzing. However, the more data are loaded when starting the server, the slower the fuzzing execution speed would become. For `Redis`, the `rdb` file we use for fuzzing is about 3.8 KB, and we have tested that it does not slow down the fuzzing execution speed too much, and also supports good code coverage.

### The fuzzer

DAFuzz is implemented based on Superion (*Wang et al., 2019*), which is further based on AFL (*Zalewski, 2017*). We implement the data-aware and semantics-aware input generation methods using C++. We also use ANTLR (ANother Tool for Language

Recognition) 4 (v4.9.3) for recognizing the ASTs of seeds and making syntax-aware seed mutations, though with new grammars. The mutation and generation methods are implemented using a predefined function interface, which follows the same framework utilized by Superion. AFL++ (*Fioraldi et al., 2020*) also adopts similar frameworks for custom mutations. As a result, switching implementations becomes effortless by loading different dynamic libraries (*i.e.*, different `.so` files). Superion disables the havoc/splice stage by default during fuzzing, but we find such a stage is useful for in-memory data stores and we re-enable it. We also implement a mode for testing multi-dimensional fuzzing as we will explain in a later "The Effect of Multi-Dimensional Fuzzing" section.

## EVALUATION

In this section, we evaluate our DAFuzz prototype, aiming to answer the following research questions.

- Does DAFuzz improve the code coverage when fuzzing in-memory data stores?
- Can DAFuzz find more vulnerabilities or find vulnerabilities more quickly?

### Experiment setup

#### Programs

We use the two most popular in-memory data stores, `Redis` and `Memcached`, for experiments. For performance comparison, we use `Redis` unstable (master) branch with the last commit #5460c10 (2022/1/3), and `Memcached` 1.6.13.

#### Baseline fuzzers

We select several state-of-the-art fuzzers for comparisons, including AFL (*Zalewski, 2017*), Superion (*Wang et al., 2019*), AFLNET (*Pham, Böhme & Roychoudhury, 2020*), and AFL++ (*Fioraldi et al., 2020*). The first one is the latest AFL (v2.52b) (*Zalewski, 2017*). Superion (*Wang et al., 2019*) is a fuzzer that supports grammar-aware (mainly syntax-aware) mutation. We update it with the grammar of commands and network protocols of the two data stores. AFLNET (*Pham, Böhme & Roychoudhury, 2020*) is selected because it is specially designed for fuzzing network protocols, and we update it with the network protocols of the two data stores as well. AFL++ (*Fioraldi et al., 2020*) (v4.00c) is the successor of AFL and has many improvements like better seed scheduling, more mutators, and faster instrumentation. For all the fuzzers we use their default parameters. All fuzzers are explicitly configured to skip the deterministic stage (*i.e.*, with "-d" option), except AFLNET and AFL++, which have disabled the stage by default (*Pham, Böhme & Roychoudhury, 2020*; *Fioraldi et al., 2020*). However, since Superion needs deterministic fuzzing for grammar-aware trimming and enhanced dictionary-based mutation (*Wang et al., 2019*), we still do deterministic fuzzing for seeds with a probability, even the "-d" option is set. We enable the probability for AFL, Superion, and DAFuzz for a fair comparison, but do not enable it for AFLNET and AFL++, which have disabled the deterministic stage by default. We set the probability to be 0.05 since we find it makes a good balance on different stages.

## Fuzzing parameters

For a fair comparison, we provide all the fuzzers with the same dictionary, the same initial seeds, and the same constructed data set (*i.e.*, constructed as described in the "Data Contruction" section). The dictionary contains the names of data store commands and the names of data items in the data set. The program parameter of `Redis` is like `./redis-server --dbfilename data.rdb --bind 127.0.0.1 --protected-mode no`, where `data.rdb` is the file contains the constructed data set, and the program parameter of `Memcached` is like `./memcached`, and the constructed data are stored in initial seeds. We use a faster de-socketing tool, `desockmulti` (*Zeng et al., 2020*), for fuzzing both programs, since they communicate with their clients through sockets but not files or `stdin`, except for AFLNET which directly communicates with the PUTs through ordinary sockets.

## Platform

The experiments are conducted on a server with 2 Intel(R) Xeon(R) CPU E5-2640 v4 @ 2.40 GHz processors, 64 GB RAM memory, and with 64-bit Ubuntu 20.04 LTS as server operating system. Each case lasts for 24 h on a single core and is repeated five times if not explicitly stated, for reducing the randomness of fuzzing (*Klees et al., 2018*). It is worth noting that the use of a 24-h run is a popular setting for fuzz testing, particularly after the publication of *Klees et al. (2018)*'s article. This is because some fuzzers may start slow and bugs often reside in certain parts of the program (*Klees et al., 2018*). Therefore, longer fuzzing runs are considered fairer and can provide a more accurate representation of performance trends in real-world scenarios (*Klees et al., 2018*).

## The comparison of code coverage

### Edge coverage

Edge coverage (*i.e.*, branch coverage) is used here since it is one of the most widely used coverage metrics now (*Lemieux & Sen, 2018*; *Fioraldi et al., 2020*; *Wang, Song & Yin, 2021*; *Metzman et al., 2021*; *Fioraldi, Maier & Balzarotti, 2022*), and the edge coverage growth of different fuzzers is shown in Figs. 4 and 5, for `Redis` and `Memcached` respectively. We can see that DAFuzz outperforms all other fuzzers in both programs. AFL, Superion, and AFL++ perform similarly and are in the second tier, while AFLNET performs considerably worse. This is mainly because the execution speed of AFLNET is slow (*e.g.*, less than 20 execs/s *vs*. over 100 execs/s for other fuzzers), which is a known problem (*Zeng et al., 2020*; *Schumilo et al., 2022*) since it feeds inputs to PUT through ordinary INET sockets but not faster UNIX sockets (*Zeng et al., 2020*). Superion outperforms its base fuzzer AFL in `Redis` while performs similarly in `Memcached`, which suggests that syntax-aware mutation (the "Syntax-Aware Mutation") may be only useful in some cases. In addition, DAFuzz performs much better than its base fuzzer Superion in both programs, which suggests that data-aware and semantics-aware input generation module (the "Data-aware and Semantics-aware Input Generation" section) could further boost the capability of the fuzzer, since the input generation module is the only difference between them in the experiment.

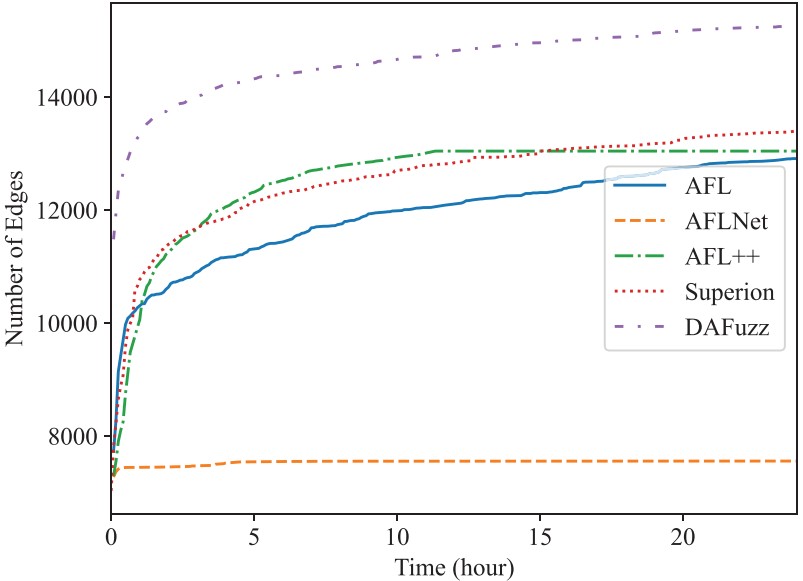

**Figure 4** **The edge coverage growth discovered by different fuzzers for** `Redis`.

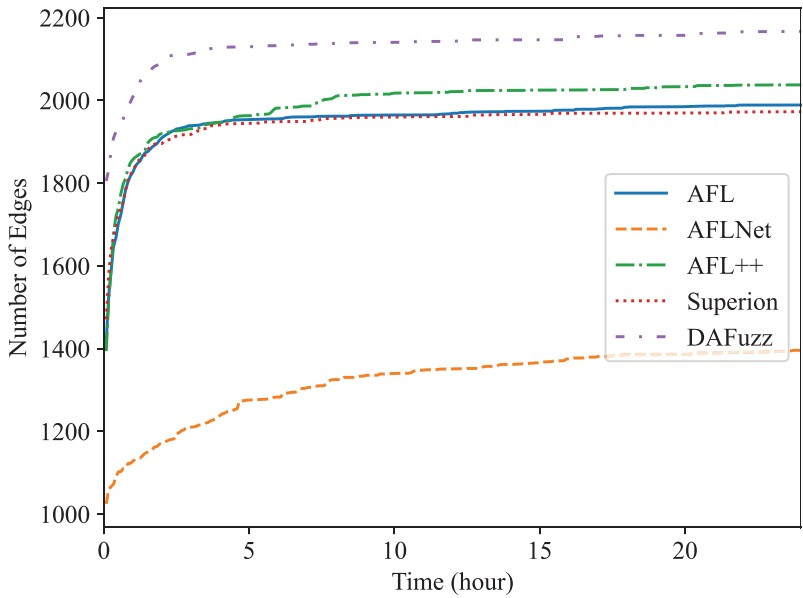

**Figure 5** **The edge coverage growth discovered by different fuzzers for** `Memcached`.

The final numbers of edges discovered by different fuzzers in 24 h are shown in Table 2 for quantitative comparison. DAFuzz discovers 17,424 edges in total in the two programs, which are 17%, 95%, 16%, and 13% more than AFL, AFLNᴇᴛ, AFL++, and Superion, respectively. We further calculate the time DAFuzz needs to discover the same number of edges other fuzzers discover in 24 h and list the time and improvements of DAFuzz over others in Table 3. The number of edges is collected every 5 min. We can see that DAFuzz

**Table 2 Average numbers of edges discovered after 24 h, with the ratios in brackets representing how many more edges DAFuzz discovers than them.**

| Program | AFL | AFLNET | AFL++ | Superion | DAFuzz |
|---------|-----|--------|-------|----------|--------|
| Redis | 12,913 (+18%) | 7,555 (+102%) | 13,043 (+17%) | 13,398 (+14%) | 15,257 |
| Memcached | 1,989 (+9%) | 1,396 (+55%) | 2,037 (+6%) | 1,973 (+10%) | 2,167 |
| Total | 14,902 (+17%) | 8,951 (+95%) | 15,080 (+16%) | 15,371 (+13%) | 17,424 |

**Table 3 The time needed by DAFuzz to discover the same numbers of edges discovered by other fuzzers in 24 h (measured every 5 min), with the improvements DAFuzz over them placed in brackets.**

| Program | AFL | AFLNET | AFL++ | Superion |
|---------|-----|--------|-------|----------|
| Redis | 40 min (36×) | <5 min (288×) | 45 min (32×) | 1 h 10 min (21×) |
| Memcached | 55 min (26×) | <5 min (288×) | 1 h 15 min (19×) | 50 min (29×) |

**Table 4 The $p$ values on the numbers of edges discovered by DAFuzz and other fuzzers.**

| Program | $p1$ | $p2$ | $p3$ | $p4$ |
|---------|------|------|------|------|
| Redis | 0.008 | 0.008 | 0.008 | 0.008 |
| Memcached | 0.008 | 0.008 | 0.008 | 0.008 |

needs at most 1 h and 15 min to discover the same number of edges any other fuzzer discovers in 24 h, which means DAFuzz is at least 19× faster in edge discovery.

*Statistical analysis*

We also use the $p$ value of Mann Whitney U-test to decide whether there is a statistically significant difference between two sets of results, as suggested in *Klees et al. (2018)*. The $p$ values are shown in Table 4, where $p1$, $p2$, $p3$, $p4$, and $p5$ represent the differences between DAFuzz and the other four fuzzers, AFL, AFLNET, AFL++, and Superion respectively. $p$ value is less than 0.05 means the difference is statistically significant. Here all the $p$ values are less than 0.01, which means the code coverage of DAFuzz is different from the code coverage of other fuzzers significantly.

## The comparison of unique vulnerability discovery

All the fuzzers only discover crashes in `Redis` during performance comparison. We use AddressSanitizer (*Serebryany et al., 2012*) to rebuild the program, run it against the inputs that cause crashes, and manually remove duplicated vulnerabilities. Eventually, we confirm that two unique vulnerabilities (one segmentation violation and one stack buffer overflow) are discovered by all fuzzers except AFLNET, which does not discover any vulnerabilities. We report the two vulnerabilities to developers in GitHub issue #10070 (https://github. com/redis/redis/issues/10070) and issue #10076 (https://github.com/redis/redis/issues/ 10076) respectively. Both of them have been confirmed and fixed by the developers. We

**Table 5 Average time needed for the vulnerability discovery, with the improvements of DAFuzz over them in brackets.**

| Unique vulnerability | AFL | AFLNᴇᴛ | AFL++ | Superion | DAFuzz |
|---|---|---|---|---|---|
| Redis issue #10070 (segmentation violation) | 12 min | >24 h | 8 min | 16 min | 19 min |
| Redis issue #10076 (stack buffer overflow) | 3 h 43 min | >24 h | 3 h 29 min | 59 min | 9 min |
| Total | 3 h 55 min (8.4×) | >24 h (>51.4×) | 3 h 37 min (7.8×) | 1 h 15 min (2.7×) | 28 min |

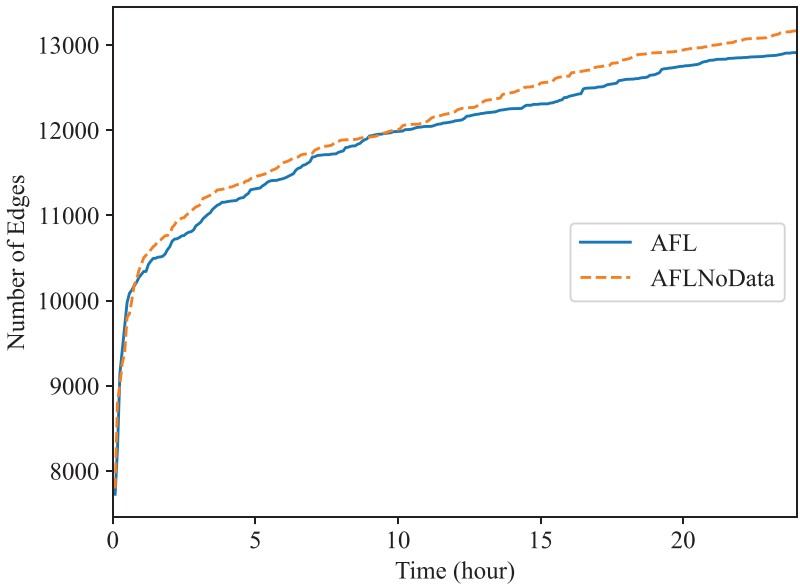

**Figure 6 Fuzzing Redis with or without constructed data.**

will describe their details later in the "Vulnerabilities Discovered" section, and focus on the comparison of vulnerability discovery speed here. Table 5 shows the average time needed for each fuzzer to discover the two vulnerabilities. DAFuzz only needs 28 min on average to find both of them, which is 2.7× faster than Superion (1 h 15 min), 7.8× faster than AFL++ (3 h 37 min), and 8.4× faster than AFL (3 h 55 min).

## The effect of using constructed data only

In this subsection, we check the effect of the data construction module alone. In the "The Comparison of Code Coverage" section, since all fuzzers are provided with the constructed data as mentioned before, the comparison between Superion and AFL illustrates the effect of syntax-aware mutation module (Superion is based on AFL but with this extra mutation module), and the comparison between DAFuzz and Superion illustrate the effect of data-aware and semantics-aware input generation module (DAFuzz is based on Superion but with this extra generation module). It may be interesting to know the effect of using the data constructed alone, and we use the base fuzzer AFL as an example. Here, one AFL fuzzer uses the data constructed (marked as AFL), another AFL fuzzer has no data provided (marked as AFLNoData), and the result is shown in Figs. 6 and 7. The code

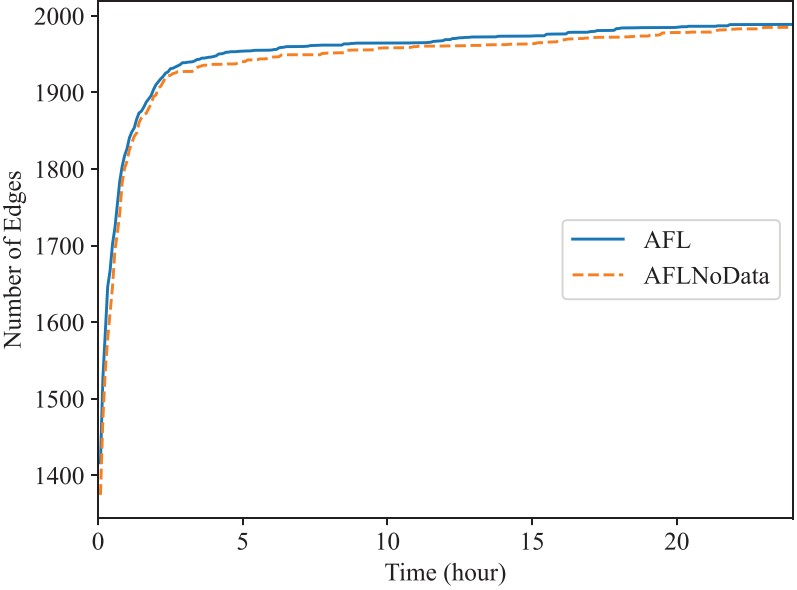

**Figure 7 Fuzzing Memcached with or without constructed data.**

**Table 6 The *p* values on with or without constructed data.**

| Program | *p*1 |
|---|---|
| Redis | 0.310 |
| Memcached | 1.000 |

coverage of AFL and AFLNoData is close in the figures. We confirm that by also using the *p* value of Mann Whitney U-test, and showing the *p* values in Table 6. Both *p* values are larger than 0.05, which means that there are no statistically significant differences between the two fuzzers. We think this may have several reasons. First, although code lines executed only when the constructed data are available are usually critical, they may be only a small portion of the whole code base. Second, code coverage may not reflect program state changes caused by satisfying different data conditions, since code coverage only cares about code paths newly being discovered but not variables getting new values (*Aschermann et al., 2020*; *Fioraldi, Elia & Balzarotti, 2021*). Third, loading extra data when starting the PUT consumes more time. For example, the average executions per second of AFL and AFLNoData are 134 and 137 exec/s for `Redis`, respectively. Thus, we can know that only using the constructed data may not improve the overall fuzzing efficiency, and other modules of DAFuzz are needed as well.

## The effect of multi-dimensional fuzzing

Recently multi-dimensional fuzzing (*Xu et al., 2019*; *Schumilo et al., 2020*; *Zou et al., 2021*; *Xie et al., 2022*) is proposed for fuzzing programs expecting two or more types of inputs at the same time. For example, two types of inputs, disk-image input and system-call input,

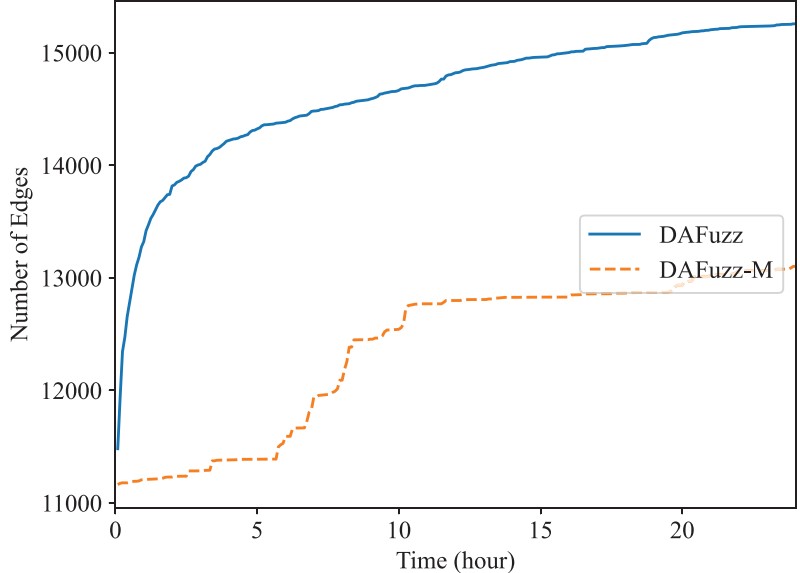

**Figure 8 Fuzzing Redis using or not using multi-dimensional fuzzing.**

are used for fuzzing file systems (*Xu et al., 2019*). In DAFuzz, the constructed data file provided for fuzzing may be considered as another type of input in addition to the command type input. Thus, we also test the effect of multi-dimensional fuzzing in DAFuzz, with a special mode we named DAFuzz-M. In DAFuzz-M mode, a seed contains two parts: the command input part and the data input part (although for simple implementation only the *index* of the data input part is actually stored in the seed, and the content of data input is stored in a queue similar to the original seed queue). During fuzzing, we follow the method in *Xu et al. (2019)* by first mutating the data input part and using the original command input part of the seed to fuzz, and then mutating the command input part and using the original data input part to fuzz. We only test multi-dimensional fuzzing with `Redis`, since `Memcached` actually is fuzzed using one type of inputs (the constructed data for `Memcached` are also loaded by using commands inserted in the front of seeds as we explained before). The result is shown in Fig. 8. However, DAFuzz-M has a lower code coverage than DAFuzz. We find it is mainly because the `rdb` file used as data input has a highly structured format (https://github.com/sripathikrishnan/redis-rdb-tools/blob/master/docs/RDB_File_Format.textile), and mutating such file could easily fail the checks during parsing (*e.g.*, inserting a single byte may cause a later length field to read at a wrong position and get an invalidly large value). In addition, the command inputs of DAFuzz may also modify the data of the data store since the commands operate on the data as well, which makes the advantage of introducing the data inputs in another dimension not apparent.

## Vulnerabilities discovered

In the following, we briefly introduce the two vulnerabilities discovered during the performance comparison experiments, and another two vulnerabilities we discovered

**Table 7 The vulnerabilities newly found by us.**

| Program | Version | Vulnerability type | Github issue no. | Status |
|---------|---------|--------------------|--------------------|--------|
| Redis | Unstable branch | Segmentation violation | Redis issue #10070 | Fixed in 7.0 |
| Redis | Unstable branch | Stack buffer overflow | Redis issue #10076 | Fixed in 7.0 |
| Redis | 6.2.1 | Segmentation violation | Redis issue #8712 | Fixed in 6.2.3 |
| Memcached | 1.6.9 | Segmentation violation | Memcached issue #779 | Fixed in 1.6.10 |

previously with DAFuzz (we used DAFuzz to fuzz `Redis` 6.2.1 and `Memcached` 1.6.9 for finding bugs only and did not compare with other fuzzers). These vulnerabilities may be used for exploitation or DoS (Denial-of-Service) attack. They are shown in Table 7 and further explained below.

*Wrongly processing commands containing "|"* (redis issue #10070). `Redis` plans to treat subcommands as commands in v7.0, which would allow having different ACL (access control list) categories for subcommands. For example, "CONFIG GET" is allowed but not "CONFIG SET", and users may send commands like "ACL SETUSER test +CONFIG| GET" to configure that. However, the processing codes added for splitting the command with "|" make some commands that contain "|" (*e.g.*, "scard|set1") wrongly pass the built-in "ERR unknown command" check, and crash in the function `addReplySubcommandSyntaxError`. The developers fix the problem by introducing a new function `isContainerCommandBySds` in `server.c` to check whether a command is a container command (*e.g.*, having subcommands) and reject the command early if it is not.

*Unexpected commands sent from replicas are not filtered* (redis issue #8712 (https:// github.com/redis/redis/issues/8712), and #10076). `Redis` supports high availability and failover with replication, and it only allows replicas to send limited commands like REPLCONF and PING to the master. We find in `Redis` 6.2.1 that after a replica sends a PSYNC and a FAILOVER command (meaning starting partial synchronization and coordinated failover between the replica and the master), the replica sends other commands like SET would cause segmentation violation in the master. Developers fix the issue by rejecting commands from replicas that interact with the keyspace of the master in function `processCommand` of `server.c`. However, the fix is incomplete unfortunately. In the unstable branch, we later find a stack buffer overflow that is triggered by a PSYNC command with a following SLOWLOG command. The developers further add code in the function `addReplyDeferredLen` of `networking.c` to disconnect replicas that send commands on the replication link that cause replies to be generated.

*External storage is not checked for stats command* (memcached issue #779, https://github. com/memcached/memcached/issues/779). *Extstore* in `Memcached` is to reduce memory footprint by leaving the hash table and keys in memory and moving values to external storage (usually flash). However, in Memcached 1.6.9, a segmentation violation would be triggered when "stats extstore" command is executed. This is because *extstore* is on by default but the server needs to be started with -o option like -o ext_path=/path/to/a/

`datafile:5G`. Otherwise, an *extstore* value is null but would be used if that "stats extstore" command is executed. Developers fix it simply by adding a null check when processing the "stats" command in the function `process_extstore_stats` of `storage.c`.

# RELATED WORK

## Coverage-guided fuzzing

Starting with the invention of AFL in 2007 (*Zalewski, 2017*), coverage-guided fuzzing (CGF) is one of the most popular fuzzing technologies (*Manes et al., 2019*; *Li et al., 2021*; *Zhu et al., 2022*). CGF fuzzers are typical grey-box fuzzers, since they only collect lightweight coverage information of inputs, and use that to guide the fuzzers to gradually explore the state space of the PUT (*Zalewski, 2017*; *Honggfuzz, 2023*; *libFuzzer, 2023*; *Fioraldi et al., 2020*; *Fioraldi, Maier & Balzarotti, 2022*). In contrast, black-box fuzzers do not know any internal execution information of the PUT (though they may know the grammar of inputs like Boofuzz (*Boofuzz, 2023*)), and white-box fuzzers know the most detailed information of the PUT (*e.g.*, through symbolic execution) (*Manes et al., 2019*). Moreover, CGF fuzzers usually are *mutation-based* since they mutate seeds to get new inputs, but there are CGF fuzzers that are *generation-based* as well, which generate new inputs from scratch by using the grammar (or say, model) information (*Manes et al., 2019*; *Li et al., 2021*; *Zhu et al., 2022*). For example, syzkaller (*Google, 2015*) could mutate existing sequences of syscalls (*i.e.*, seeds) and generate new sequences of syscalls at the same time during fuzzing. DAFuzz follows the same approach. CGF fuzzers have successfully been used to discover many vulnerabilities (*Zalewski, 2017*; *Google Security Team, 2018*), and become popular in both the security industry (*Zalewski, 2017*; *Honggfuzz, 2023*; *libFuzzer, 2023*; *Fioraldi et al., 2020*; *Google Security Team, 2018*) and academia (*Böhme, Pham & Roychoudhury, 2016*; *Lemieux & Sen, 2018*; *Gan et al., 2018*; *Lyu et al., 2019*; *Pham, Böhme & Roychoudhury, 2020*; *Aschermann et al., 2020*; *Yue et al., 2020*; *Wang, Song & Yin, 2021*; *Lin et al., 2022*; *Fioraldi, Maier & Balzarotti, 2022*).

There are two kinds of fuzzing techniques that are related to the data-aware fuzzing technique proposed here, but are actually different. One is fuzzing with the aid of data-flow information (*Wang et al., 2010*; *Rawat et al., 2017*; *Chen & Chen, 2018*; *Aschermann et al., 2019b*; *Gan et al., 2020*; *Mantovani, Fioraldi & Balzarotti, 2022*). CGF fuzzing usually uses control-flow information only, *e.g.*, coverage based on the edges of the control flow graph (CFG). However, by using techniques like dynamic taint analysis, fuzzers could know information like which bytes of the inputs are used in branch instructions. Such information could further guide the fuzzer to bypass magic-byte and checksum checks (*Wang et al., 2010*; *Rawat et al., 2017*; *Aschermann et al., 2019b*), mutate seeds more efficiently (*Chen & Chen, 2018*; *Gan et al., 2020*; *Mantovani, Fioraldi & Balzarotti, 2022*), or use as another interest feedback besides coverage feedback (*Mantovani, Fioraldi & Balzarotti, 2022*). Generally, such data-flow information extra collected is about the data in the input, while in DAFuzz the data concerned are stored in the in-memory data store (*i.e.*, the PUT).

Another related technique is the recently proposed state-aware fuzzing (*Aschermann et al., 2020*; *Fioraldi, Elia & Balzarotti, 2021*; *Ba et al., 2022*). It usually considers the whole

program state space to be divided into different regions by different values of some important variables. Such important variables could be variables representing the states of protocols (*Aschermann et al., 2020*; *Ba et al., 2022*), or variables that stay the same values for most inputs but change to other values for some inputs (*Fioraldi, Elia & Balzarotti, 2021*). The CGF fuzzers detect the important variables either manually (*Aschermann et al., 2020*) or automatically (*Fioraldi, Elia & Balzarotti, 2021*; *Ba et al., 2022*) and use them together to guide the exploration of state space during fuzzing. Different from DAFuzz, state-aware fuzzing also usually concerns the important variables related to the input but not the data stored in the program. In addition, the important variables usually are variables related to the switching of execution paths, but DAFuzz directly concerns the data and may work even if no such variables are explicitly defined.

### Grammar-aware fuzzing

There are black-box (*Peach Tech, 2020*; *Boofuzz, 2023*), grey-box (*Pham et al., 2021*), and white-box (*Godefroid, Kiezun & Levin, 2008*) fuzzers that support grammar-aware fuzzing. For example, black-box fuzzers like Peach (*Peach Tech, 2020*) and Boofuzz (*Boofuzz, 2023*) use XML configuration files or code to define the format of inputs to generate inputs, and white-box fuzzers like (*Godefroid, Kiezun & Levin, 2008*) may generate extra constraints using the grammar of inputs. For grey-box fuzzers, researchers have found that CGF fuzzers do not perform well for programs expecting structural inputs, such as JavaScript engines, XML parsers, *etc*., because when mutating seeds to get new inputs, it is easy to get invalid inputs (*Wang et al., 2019*), which cannot pass the syntax and semantics checks early in the program execution. Thus, grammar-aware CGF is proposed.

Grammar-aware CGF could roughly be divided into syntax-aware (*Pham et al., 2021*; *Wang et al., 2019*; *Padhye et al., 2019*; *Aschermann et al., 2019a*; *Pham, Böhme & Roychoudhury, 2020*; *Salls et al., 2021*), and semantics-aware fuzzing (*Han, Oh & Cha, 2019*; *Park et al., 2020*; *He et al., 2021*). Syntax-aware fuzzing tries to ensure the inputs mutated from seeds are still in the correct format. For example, they may convert seeds into abstract syntax trees (ASTs) and mutate at the tree node level (*i.e*., tree-based mutation) to ensure the inputs serialized from mutated ASTs still have correct JavaScript statements (*Wang et al., 2019*). Semantics-aware fuzzing tries to further ensure the inputs mutated from seeds have valid semantics. For example, they may inspect the JavaScript code or runtime errors to ensure the variables are defined before use (*Han, Oh & Cha, 2019*; *Park et al., 2020*; *He et al., 2021*). DAFuzz adopts the same tree-based syntax-aware mutation (*Wang et al., 2019*), which is enough since in-memory data stores usually could use data variables without defining them first. However, different from the existing work, DAFuzz also generates data-aware and semantics-aware new inputs, which is to ensure that different data items are correctly referred to in the inputs.

### Data-related program fuzzing

There are researchers focusing on fuzzing data-related programs. For in-memory data store, Google security researchers fuzzed `Redis` but only used existing ordinary fuzzers like AFL (*Google Information Security Engineering Team, 2020*). Another kind of data-

related program is SQL database, and different special black-box fuzzers (*Seltenreich, Tang & Mullender, 2022*; *Guo, 2017*; *Rigger, 2023*; *Rigger & Su, 2020*) and grey-box fuzzers (*Zhong et al., 2020*; *Wang et al., 2021*; *Liang, Liu & Hu, 2022*) have been developed. At first, fuzzers mainly focus on keeping the generated or mutated SQL statements syntactically valid (*Seltenreich, Tang & Mullender, 2022*; *Guo, 2017*). Recently, fuzzers also try to ensure the statements are semantically valid (*Rigger, 2023*; *Rigger & Su, 2020*; *Zhong et al., 2020*; *Wang et al., 2021*; *Liang, Liu & Hu, 2022*). For example, they ensure that the used tables are created first and the mentioned columns still exist (*Zhong et al., 2020*; *Liang, Liu & Hu, 2022*). Several recent pieces of research focus on detecting logic bugs but not traditional crashes or assert failures (*Rigger, 2023*; *Rigger & Su, 2020*; *Liang, Liu & Hu, 2022*). However, all these previous works do not intentionally create different kinds of data as DAFuzz does for providing the required data, and all these fuzzers are tightly bound to the SQL language and cannot work with in-memory data stores that use other languages.

## CONCLUSION

To exercise the code paths of in-memory data stores that require different data, we presented a new fuzzing approach DAFuzz. DAFuzz could not only generate inputs that are syntactically and semantically valid but also use different data correctly. In addition, DAFuzz adopts the state-of-the-art tree-based mutation method as well. The comparisons with other state-of-the-art fuzzers like AFL, AFL++, Superion, and AFLNET in two popular in-memory data stores Redis and Memcached showed that DAFuzz could discover 13~95% more edges, or discover the same number of edges at least 19× faster. Furthermore, DAFuzz found the same vulnerabilities but over 2.7× faster. We newly found three vulnerabilities in Redis and one vulnerability in Memcached, and reported them to developers. All vulnerabilities have been acknowledged and fixed. We also believe that the concept of data-aware fuzzing can be applied to other in-memory data stores, such as Dragonfly. Furthermore, it has the potential to be employed in fuzzing other software systems that exhibit behavior dependent on different data conditions.

### Funding
This work was supported by the Zhejiang Provincial Natural Science Foundation of China under Grant No. LY22F020022, the National Natural Science Foundation of China under Grant No. 61902098, the Key Research Project of Zhejiang Province, China under Grant No. 2023C01025, and the "Pioneer" and "Leading Goose" R&D Program of Zhejiang under Grant No. 2023C03203. There was no additional external funding received for this study. The funders had no role in study design, data collection and analysis, decision to publish, or preparation of the manuscript.

### Grant Disclosures
The following grant information was disclosed by the authors:
Zhejiang Provincial Natural Science Foundation of China: LY22F020022.

National Natural Science Foundation of China: 61902098.
Key Research Project of Zhejiang Province, China: 2023C01025.
"Pioneer" and "Leading Goose" R&D Program of Zhejiang: 2023C03203.

## Competing Interests

The authors declare that they have no competing interests.

## Author Contributions

- Yingpei Zeng conceived and designed the experiments, performed the experiments, analyzed the data, performed the computation work, prepared figures and/or tables, authored or reviewed drafts of the article, and approved the final draft.
- Fengming Zhu performed the experiments, analyzed the data, performed the computation work, prepared figures and/or tables, authored or reviewed drafts of the article, and approved the final draft.
- Siyi Zhang performed the experiments, analyzed the data, performed the computation work, prepared figures and/or tables, and approved the final draft.
- Yu Yang performed the experiments, analyzed the data, prepared figures and/or tables, and approved the final draft.
- Siyu Yi performed the experiments, analyzed the data, prepared figures and/or tables, and approved the final draft.
- Yufan Pan performed the experiments, analyzed the data, prepared figures and/or tables, and approved the final draft.
- Guojie Xie conceived and designed the experiments, authored or reviewed drafts of the article, and approved the final draft.
- Ting Wu conceived and designed the experiments, authored or reviewed drafts of the article, and approved the final draft.

## Data Availability

The code is available in the Supplemental File.

## Supplemental Information

Supplemental information for this article can be found online at http://dx.doi.org/10.7717/peerj-cs.1592#supplemental-information.

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
