# Peer review of "DAFuzz: data-aware fuzzing of in-memory data stores"

_PeerJ Computer Science, doi:10.7717/peerj-cs.1592_

## Round 0.1 · original submission · Minor Revisions

The reviewers have recommended minor changes to the submitted manuscript. Please see the reviewers' comments and revise the manuscript accordingly.

Also, consider the following changes in the revised version of the paper:
- Add a related work summary table comparing the core available approaches in terms of the multiple aspects.
- The discussion of the related techniques is very brief. Extend the discussion with more detail about the methods' working semantics along with the limitations.
- Extend the implementation section with more details about the implementation.

Reviewer 1 ·

Basic reporting

The paper is written in clear professional English.
The literature review is comprehensive.
The results reporting is professional.

Experimental design

The experiments and results looks valid.

Validity of the findings

The experiments and results looks valid.
The proposed method is used to successfully uncover real vulnerabilities in Redis and Memcached.
A weakness of the proposed method is that it seems it's limited to in-memory data-stores. Can it be generalized to other softwares whose behavior depend on different dataset/states?

Additional comments

I'm not an expert on fuzzing so the review is mostly about overall writing.

Cite this review as
Anonymous Reviewer (2023) Peer Review #1 of "DAFuzz: data-aware fuzzing of in-memory data stores (v0.1)". PeerJ Computer Science

Reviewer 2 ·

Basic reporting

A well written paper. The research is ambitious and correctly presented.

With reference to the flow/ readability of the research it is not very welcoming for the reader who is just entering into the domain. This is a point the authors can consider for improving the overall readability.

Having said this I will recommend that the literature review be extended to incorporate a breadth of knowledge.

Also highlight the known downsides of using fuzzing. The vulnerabilities being discovered through the method can also be listed rather than being placed within the text.

Experimental design

The experimental design is correct and appropriate to the research.

Although I will ask that the authors have mentioned a pretty high time requirement which is expected from fuzzing based implementation. This point can be discussed further and the underlying reason highlighted.

Validity of the findings

The comparison of the work in textual form again does not make an impression. Authors can explore how comparative analysis can be improved for readability.

Additional comments

Minor Revisions suggested.

Cite this review as
Anonymous Reviewer (2023) Peer Review #2 of "DAFuzz: data-aware fuzzing of in-memory data stores (v0.1)". PeerJ Computer Science

---

## Round 0.2 · accepted · Accept

Congratulations, the revised version is satisfactory and recommended for publication.

Reviewer 2 ·

Basic reporting

The style and language of reporting has been improved. Changes suggested by the reviewers have been incorporated.

Experimental design

There was no issue in the experimental design. I asked for some clarifications and improvements which have been incorporated.

Validity of the findings

No issue seen.

Additional comments

All my concerns have been addressed.

Cite this review as
Anonymous Reviewer (2023) Peer Review #2 of "DAFuzz: data-aware fuzzing of in-memory data stores (v0.2)". PeerJ Computer Science